# Changes in Lymphocyte Subpopulations after Remdesivir Therapy for COVID-19: A Brief Report

**DOI:** 10.3390/ijms241914973

**Published:** 2023-10-07

**Authors:** Rossella Cianci, Maria Grazia Massaro, Elisabetta De Santis, Beatrice Totti, Antonio Gasbarrini, Giovanni Gambassi, Vincenzo Giambra

**Affiliations:** 1Department of Translational Medicine and Surgery, Università Cattolica del Sacro Cuore, 00168 Rome, Italy; mg.massaro92@gmail.com (M.G.M.); antonio.gasbarrini@unicatt.it (A.G.); giovanni.gambassi@unicatt.it (G.G.); 2Fondazione Policlinico Universitario A. Gemelli, Istituto di Ricovero e Cura a Carattere Scientifico (IRCCS), 00168 Rome, Italy; 3Institute for Stem Cell Biology, Regenerative Medicine and Innovative Therapies (ISBReMIT), Fondazione IRCCS “Casa Sollievo della Sofferenza”, 71013 San Giovanni Rotondo, Italy; e.desantis@operapadrepio.it (E.D.S.); b.totti@operapadrepio.it (B.T.); v.giambra@operapadrepio.it (V.G.)

**Keywords:** COVID-19 disease, lymphocyte subpopulations, remdesivir, immune modulation

## Abstract

Remdesivir (RDV) has demonstrated clinical benefit in hospitalized COronaVIrus Disease (COVID)-19 patients. The objective of this brief report was to assess a possible correlation between RDV therapy and the variation in lymphocyte subpopulations. We retrospectively studied 43 hospitalized COVID-19 patients: 30 men and 13 women (mean age 69.3 ± 15 years); 9/43 had received RDV therapy. Six patients had no need for oxygen (severity group 0); 22 were on oxygen treatment with a fraction of inspired oxygen (FiO_2_) ≤ 50% (group 1); 7 on not-invasive ventilation (group 2); 3 on invasive mechanical ventilation (group 3); and 5 had died (group 4). Cytofluorimetric assessment of lymphocyte subpopulations showed substantial changes after RDV therapy: B lymphocytes and plasmablasts were significantly increased (*p* = 0.002 and *p* = 0.08, respectively). Cytotoxic T lymphocytes showed a robust reduction (*p* = 0.008). No changes were observed in CD4^+^-T cells and natural killers (NKs). There was a significant reduction in regulatory T cells (Tregs) (*p* = 0.02) and a significant increase in circulating monocytes (*p* = 0.03). Stratifying by disease severity, after RDV therapy, patients with severity 0–2 had significantly higher B lymphocyte and monocyte counts and lower memory and effector cytotoxic T cell counts. Instead, patients with severity 3–4 had significantly higher plasmablast and lower memory T cell counts. No significant differences for CD4^+^-T cells, Tregs, and NKs were observed. Our brief report showed substantial changes in the lymphocyte subpopulations analyzed between patients who did not receive RDV therapy and those after RDV treatment. Despite the small sample size, due to the retrospective nature of this brief report, the substantial changes in lymphocyte subpopulations reported could lead to speculation on the role of RDV treatment both on immune responses against the virus and on the possible downregulation of the cytokine storm observed in patients with more severe disease.

## 1. Introduction

Remdesivir (RDV) (GS-5734) is a nucleotide prodrug that binds to the viral RiboNucleic Acid (RNA)-dependent RNA polymerase and inhibits Severe Acute Respiratory Syndrome CoronaVirus 2 (SARS-CoV-2) viral replication through the early end of RNA transcription [1,2]. RDV has demonstrated clinical benefit in hospitalized COronaVIrus Disease (COVID)-19 patients with moderate-to-severe disease [3,4,5]. A phase 3 clinical trial in non-hospitalized COVID-19 patients at high risk of severe disease further demonstrated that a 3-day course of RDV resulted in an 87% lower risk of hospitalization or death compared to placebo (PINETREE study) [6]. RDV inhibits the in vitro growth of many viruses, such as Filoviridae, Paramyxoviridae, Pneumoviridae, and Coronaviridae [7]. Between 2013 and 2016, RDV was used in the Ebola outbreak [8]. However, even if there have been cases of healing after treatment with RDV [9], a randomized trial conducted in 2018 demonstrated that the drug’s clinical efficacy was minimal [10]. During the 2020 COVID-19 pandemic, RDV was again studied due to its in vitro efficacy against SARS-CoV-1 and Middle-East Respiratory Syndrome CoronaVirus (MERS-CoV) [11]. Several studies have been conducted to evaluate the efficacy of the drug both in patients with severe disease and in patients with mild/moderate ones [3]. However, some authors have recently reported a reduction in the time for clinical remission, even without a difference in the duration of the swab positivity [3]. One possible explanation is the potential immunomodulatory effect of the drug [12]. The efficacy of RDV in avoiding a more serious form of the disease and in reducing its duration could depend on immune activity, as well as on an anti-viral one, as demonstrated by in vitro studies. In fact, it has not been demonstrated that patients who have received RDV become more quickly negative than patients who have not received it. Moreover, it is well known that the severity of SARS-CoV-2 infection depends, at least in part, on the inflammatory response of the host [13]. The objective of this brief report was to verify a possible correlation between RDV therapy and the variations in lymphocyte subpopulations in the hypothesis that RDV therapy modulates them to favor the antiviral response, limiting the cytokine storm phenomenon that is observed in patients with more severe disease.

## 2. Results

The lymphocyte subpopulations of COVID-19 patients showed significant changes after RDV therapy. In detail, B lymphocytes (8.37 vs. 14.48, *p* = 0.002) and plasmablasts (4.05 vs. 6.42, *p* = 0.08) showed a significant increase. Cytotoxic T lymphocytes showed a significant reduction after RDV therapy (8.08 vs. 1.74, *p* = 0.008), both for activated effector TCD8^+^ (5.51 vs. 1.41, *p* = 0.01) and for memory TCD8^+^ cells (2.46 vs. 0.31, *p* = 0.007). No changes were observed in TCD4^+^ (3.01 vs. 2.03, p ns) and NK cells. There was a significant reduction in regulatory T cells (Tregs) (0.76 vs. 0.47, *p* = 0.02) and a significant increase in circulating monocytes (2.90 vs. 4.11, *p* = 0.03). The results observed are summarized in Table 1 and in Appendix A. Stratifying the population by disease severity, patients with severity 0–2 had significantly higher B lymphocyte and monocyte levels after RDV therapy and lower memory and effector cytotoxic T lymphocyte levels. Instead, patients with severity 3–4 had significantly higher after-therapy plasmablast values while presenting significantly lower levels of memory T cells compared to patients not treated with RDV. There were no significant differences for TCD4^+^ lymphocytes, Tregs, or NKs, in relation to the severity of the disease. NK T-like cells were significantly reduced after RDV therapy, but only in patients with minor disease severity. The results of the lymphocyte subpopulation differences according to severity are summarized in Table 2. No lymphocyte population differences were observed between patients with and without interstitial pneumonia.

## 3. Discussion

T lymphocytes and natural killers mediate the immune response against viral pathogens. B lymphocytes are a critical component of humoral immunity because they produce antigen-specific neutralizing monoclonal antibodies and act as specific antigen-presenting cells, inducing the activation and differentiation of antigen-specific T cells. It is known that patients with COVID-19 receiving B lymphocyte-depleting therapy may develop a sustained SARS-CoV-2 infection. Furthermore, patients receiving monoclonal antibodies that reduce B lymphocyte counts (for example, anti-CD20 monoclonal antibodies, such as rituximab and obinutuzumab in hematologic diseases), are at increased risk of clinical complications and mortality linked to COVID-19 [14]. Our data show that after treatment with RDV, there was a significant increase in B lymphocytes, as well as in plasmablasts, which are precursors of plasma cells that produce antigen-specific antibodies. Stratifying the population by disease severity, patients with severity 0–2 showed higher B lymphocyte levels after RDV therapy, as the activation of a greater antibody response in these subjects reduced the severity of the disease. Furthermore, there was a reduction in Tregs (0.76 vs. 0.51, *p* = 0.01), which is a sign of a limitation of downregulatory mechanisms that is peculiar to this subpopulation of T lymphocytes. To corroborate the hypothesis of an immunoregulatory mechanism of RDV, there was a significant increase in circulating monocytes (2.80 vs. 4.44, *p* = 0.02) exclusively in patients who presented with less severe disease.

Monocytes help to clear cellular debris, kill pathogens, and relocate non-apoptotic cells into the blood and lymphatic system [15]. Furthermore, neutrophils, macrophages, and NK cells are the innate immune cells involved in the pathogenesis of the cytokine storm typical of COVID-19 and are associated with increased adverse disease outcomes [16]. Our data show that after treatment with RDV, there was a reduction in NK T-like cells exclusively in patients with less severe disease. NK T-like lymphocytes are effector lymphocytes for both innate and adaptive immunity; these cells are activated during infections and inflammatory states and rapidly produce high amounts of immunomodulatory cytokines [17]. In our patients, after RDV therapy, we have found reduced levels of inflammatory markers, such as IL-6, fibrinogen, D-dimer, and ferritin (see Appendix A). In the same way, in a recent study, it has been demonstrated that combination treatment with human monoclonal anti-IL-1β antibody, remdesivir, and steroids in moderate-to-severe hospitalized COVID-19 patients can reduce inflammatory markers [18].

The adaptive immune system represented by B and T cells show various effects in the cytokine storm [19]. IL-6 promotes the differentiation of naïve CD4^+^ T cells into effector and helper T cells that upregulate adaptive immunity; in addition, IL-6 also promotes differentiation into Th17 and can block CD8^+^ cytotoxic T lymphocytes by inhibiting IFN-γ secretion. Our data show that there was a reduction, although not significant, in CD4^+^ T lymphocytes (3.04 vs. 1.99, *p* ns), hypothesizing a reduction in the initiation of the cytokine storm. In addition, the levels of total CD8^+^ T lymphocytes and activated effector CD8^+^ T lymphocytes were significantly lower in patients after RDV therapy in general and particularly in those with less severe disease. We can hypothesize a lower activation of CD8^+^ T cells related to a reduction in viral load after RDV therapy. Memory CD8^+^ T lymphocytes were significantly decreased after RDV therapy in all patients for all degrees of disease severity. In a previous report of immunocompromised patients with severe COVID-19, the CD8^+^ T cell compartment showed a reduction in naïve cells and an increase in memory T lymphocytes [20].

In vivo evidence suggests that RDV is associated with reduced time to symptom resolution [21], although the impact on other clinical outcomes (such as mortality, onset of ventilation, and length of hospital stay) remain uncertain [21,22,23,24,25,26]. The latest international guidelines for the treatment of COVID-19 attest to a reduction in mortality and the need for hospitalization in patients treated with RDV, with particular attention to early administration in high-risk outpatients [27,28,29]. Studies focusing on the effect of RDV on viral outcomes, including viral load and clearance from respiratory samples, still show conflicting results [12,30,31,32]. In the hospital setting, an effect on mortality and the need for mechanical ventilation has been reported, but there are still conflicting data on the extent of efficacy of RDV in the treatment of severe COVID-19 [3,22]. Differences in results could be due to different outcomes and definitions of severity or sample size. Our brief report presents limitations: the small sample size does not permit reaching a strong statistical significance, and the lack of a specific control group. These limitations are due to the retrospective nature of this analysis. Furthermore, the participants in this study represent a widely heterogeneous group regarding disease stage, comorbidities, polypharmacology use, and vaccine administration. But these patients are representative of the real-life clinical practice, and all of them have received the same standard of care, represented in that time by steroids. Nowadays, even if vaccination is largely available, a specific and effective treatment for COVID-19 disease, especially for frail and comorbid patients, present us with new challenges to minimize the effects of new variations in viral strains.

## 4. Materials and Methods

*Patient samples:* During 2020 COVID-19 pandemic, we enrolled COVID-19 patients to perform immunological studies [33]. We have retrospectively studied the available lymphocyte subpopulations of 43 patients who developed COVID-19 during the 2020 COVID-19 pandemic. The study had previously received appropriate institutional ethical approvals by ethical committees at both the “Casa Sollievo della Sofferenza” Research Ethics Board (Approval Code: GEN-COVID, protocol number 2489/010G, 29/04/2020) and “Fondazione Policlinico Universitario Agostino Gemelli” (Approval Code: COVID-19-SGR, protocol number 0046550/20, 16/11/2020).

The total number of patients retrospectively included in this study are 43: 30 men and 13 women, mean age 69.3 ± 15 years. Regarding the severity of the disease, the hospitalized patient sample was divided as follows: 6 asymptomatic or with minor symptoms but no need for oxygen (severity group 0); 22 on oxygen treatment but with a FiO_2_ ≤ 50% (severity group 1); 7 on oxygen supplied through high-flow nasal cannula (HFNC) or receiving not-invasive ventilation (severity group 2); 3 on invasive mechanical ventilation or Extra Corporeal Membrane Oxygenation (ECMO) in an intensive care unit (severity group 3); and 5 who died during hospitalization (severity group 4). The characteristics of the study population are shown in Table 3 and Appendix A. Moreover, 15 patients had no comorbidities and no consumption of pharmacological medication at home; 20 presented with one comorbidity, represented by arterial hypertension in 13 patients; 8 patients presented with two comorbidities (including neoplasia, diabetes, arterial hypertension, obesity, ischemic heart disease). The comorbidities reported by the study population are shown in Table 4. The presence of SARS-CoV-2 had been detected by an Real Time-Polymerase Chain Reaction (RT-PCR) test from a nasal and oropharyngeal swab. All patients received 6 mg of dexamethasone. Of the 43 patients, 9 had received RDV 200 mg intravenously on the first day and 100 mg intravenously on the following three days and blood samples had been collected at the end of therapy. Of the 9 patients who received RDV, 9 (100%) had evidence of interstitial pneumonia on a CT scan; of the 34 patients who did not receive RDV, 27 (79.41%) had evidence of interstitial pneumonia.

*Isolation of peripheral blood mononuclear cells (PBMCs):* PBMCs were isolated from whole blood using Ficoll-Paque. Briefly, 6 mL of ethylenediaminetetraacetic acid (EDTA)-anticoagulated blood was diluted with an equal volume of phosphate-buffered saline, pH 7.4 (PBS), containing 0.05 M EDTA (Invitrogen, Carlsbad, CA, USA). A 12 mL volume of diluted blood was layered over 24 mL of Ficoll-Paque PLUS (GE Healthcare, Chicago, IL, USA). The gradients were centrifuged at 400× *g* for 30 min at room temperature in a swinging-bucket rotor with no braking or acceleration. The cell interface was carefully removed by pipetting and washed with PBS-EDTA by centrifugation at 250× *g* for 10 min. PBMC pellets were suspended in an ammonium chloride solution (Stemcell Technologies, Vancouver, BC, Canada) and incubated for 10 min at room temperature on a mixing platform in order to lyse any contaminating red blood cells. Isolated PBMCs were finally washed with PBS-EDTA.

*Flow cytometry assays:* To perform the indicated multiparameter flow cytometry assay, PBMCs were stained with the 13-color antibody panel in DPBS with BD Horizon Brilliant Stain Buffer (Becton Dickinson, Franklin Lakes, NJ, USA) for 20 min at room temperature following the concentrations suggested by the manufacturer. Briefly, cells were washed with DPBS and then stained for surface markers (Appendix A) in DPBS with BD Horizon Brilliant Stain Buffer (Becton Dickinson) for 20 min at room temperature. In this assay, the LIVE/DEAD™ Fixable Near-IR dead cell stain kit (cat. L34975, ThermoFisher, Waltham, MA, USA) was also included to identify the fraction of total living cells. All flow cytometry analyses were performed on a FACS Canto2 (Becton Dickinson) or MoFlo Astrios cell sorter (Beckman Coulter, Brea, CA, USA). The FlowJo (Becton Dickinson) and GraphPad-Prism 8.4.3 software were employed for the visualization and statistical analyses of data using the gating strategy reported in Table 5 and Figure 1 for the identification of different cell subpopulations of PBMCs using the considered panel (Table 5). The following cells were analyzed: B lymphocytes (CD3^−^CD19^+^HLA/DR^+^); B plasmablasts (CD3^−^CD19^+^HLA/DR^+^CD38^high^); CD8^+^ T lymphocytes (CD3^+^CD8^+^), activated effector CD8^+^ T lymphocytes (CD3^+^CD8^+^CD11b^+^); CD8^+^ memory T lymphocytes (CD3^+^CD8^+^CD11b^−^), CD4^+^ T lymphocytes (CD3^+^CD4^+^), regulatory T lymphocytes (CD3^+^CD4^+^CD127^−^); NK T-like lymphocytes (CD3^+^CD56^+^), natural killers (CD3^−^HLA/DR^−^CD56^+^), and monocytes (CD14^+^HLA/DR^+^CD16^−^). Subsequent gating identified CD3^+^ cells, followed by identification of CD4^+^CD8^−^ helper T cells and CD4^−^CD8^+^ cytotoxic T cells. Within the CD4^+^ cell population, regulatory T cells were discriminated as CD127^−^ cells. NK T cells were also detected as CD56^+^ cells in the CD3^+^ population. NK cells were recognized as CD3^−^CD56^+^CD19^−^. B cells were identified as CD3^−^CD19^+^ HLA-DR^+^ and plasmablasts were differentiated as CD38^high^ within the B cell population.

## 5. Conclusions

Our brief report showed substantial changes in the lymphocyte subpopulations analyzed between patients that did not receive RDV therapy and those after RDV treatment. Despite the small sample size due to the retrospective nature of this report, the substantial differences in lymphocyte subpopulations described could lead to speculation on the role of RDV treatment, both on the immune response against the virus and on a possible downregulation of the cytokine storm that is responsible for the increase in vascular permeability, vascular congestion, pulmonary edema, and compromised gas exchange that are observed in patients with more severe disease.

## 6. Future Directions

In the light of new data on specific antiviral therapies [34], the rapid onset of new SARS-CoV-2 variant strains, and the differences observed after RDV therapy, it is important to better understand the role of different immune cells in the fight against SARS-CoV-2 infections. Of note, the alterations in lymphocyte subpopulations are linked to different clinical conditions, especially in the presence of comorbidities affecting the immune system, such as neoplasms, diabetes, chronic inflammatory disorders, and COVID-19 disease stage. Moreover, the different functions of lymphocyte subpopulations can also vary depending on steroid treatment, which was commonly used during the COVID-19 pandemic. Even though all of patients of this report received the same dosage of steroids, substantial modifications in lymphocyte subpopulation were observed after RDV treatment. Further larger prospective studies are needed to identify the specific contribution to immune modulation by this antiviral treatment, and its link with clinical outcomes and cytokine production [35] in more homogeneous patient subgroups.

## Figures and Tables

**Figure 1 ijms-24-14973-f001:**
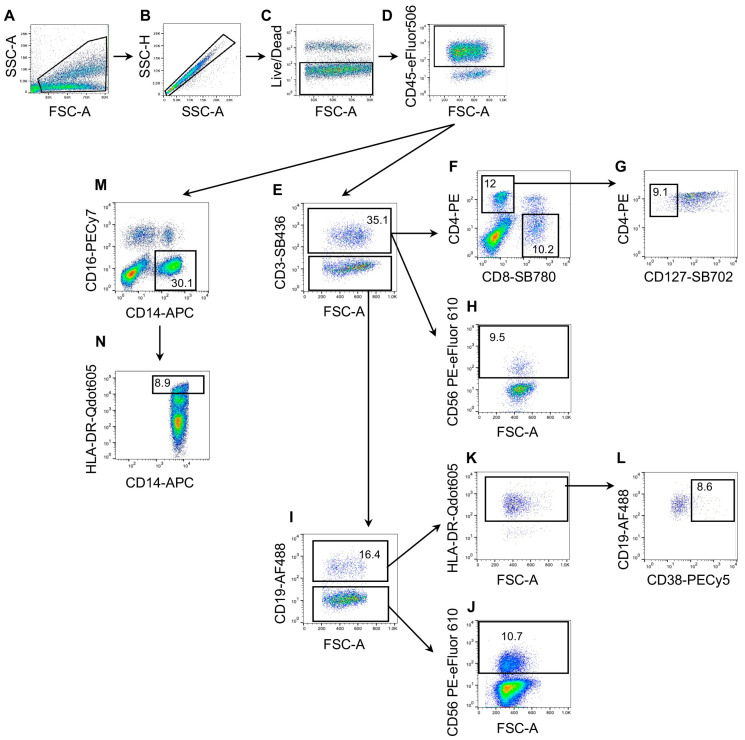
**Overview of gating strategy for identifying the different subsets in peripheral blood mononuclear cells (PBMCs) with the reported panel.** Fluorescence minus one (FMO) controls were used to set up all gates. Cell sizes (**A**) and singlets (**B**) were initially discriminated on SSC-A/FSC-A and SSC-H/SSC-A plots, respectively, followed by the exclusion of non-viable cells with Live/Dead far-red fluorescent DNA dye (**C**) and the identification of the CD45^+^ cell fraction (**D**). Subsequent gating identified CD3^+^ cells (**E**), followed by the identification of CD4^+^CD8^−^ helper T cells and CD4^−^CD8^+^ cytotoxic T cells (**F**). Within the CD4^+^ cell fraction, regulatory T cells were discriminated as CD127^−^ cells (**G**). Natural killer T-cells were also identified as CD56^+^ cells in the CD3^+^ fraction (**H**). NK cells were recognized as CD3^−^CD56^+^CD19^−^ (**I**,**J**). B cells were identified as CD3^−^CD19^+^ HLA-DR^+^ (**I**–**K**), and plasma B cells were differentiated as CD38^high^ within the subset of B cells (**L**). Finally, monocytes were discriminated as classical (CD14^+^CD16^−^), intermediate (CD14^+^CD16^+^), and non-classical (CD14^dim^CD16^+^) (**M**). Classical monocytes were subsequently subdivided as an HLA-DR^high^ cell fraction (**N**).

**Table 1 ijms-24-14973-t001:** Differences between mean values of lymphocyte subpopulations in the study population.

Cell Type	B Lymphocytes	B Plasmablasts	T CD8^+^	Activated Effector T CD8^+^	T CD8^+^ Memory	T CD4^+^	Treg	NK T-like	NK	Monocytes
**Mean values of patients not in therapy**	8.37	4.05	8.08	5.51	2.46	3.01	0.76	2.90	13.99	2.90
**Mean values of patients after RDV**	14.48	6.42	1.74	1.41	0.31	2.03	0.47	0.94	15.33	4.11
** *p* **	**0.002**	**0.008**	**0.00**	**0.01**	**0.007**	0.11	**0.02**	**0.03**	0.30	**0.03**

**Table 2 ijms-24-14973-t002:** Differences between mean values of lymphocyte subpopulations with respect to disease severity.

Cell Type	B Lymphocytes	B Plasmablasts	T CD8^+^	Activated Effector T CD8^+^	T CD8^+^ Memory	T CD4^+^	Treg	NK T-like	NK	Monocytes
**Mean values of patients not in therapy (severity 0–2)**	8.70	4.50	7.97	5.31	2.53	2.82	0.70	2.79	13.58	2.87
**Mean values of patients after RDV (severity 0–2)**	14.86	5.55	1.42	1.12	0.28	1.59	0.49	0.25	12.25	5.04
** *p* **	**0.007**	0.20	**0.02**	**0.04**	**0.03**	0.09	0.06	**0.04**	0.30	**0.004**
**Mean values of patients not in therapy (severity 3–4)**	7.88	1.86	6.75	4.19	2.49	3.45	1.00	3.13	14.39	2.48
**Mean values of patients after RDV (severity 3–4)**	13.86	7.87	2.27	1.89	0.35	2.76	0.43	2.09	20.46	2.57
** *p* **	0.10	**0.001**	0.09	0.15	**0.03**	0.31	0.05	0.33	0.09	0.44

**Table 3 ijms-24-14973-t003:** Demographic distribution and clinical features of COVID-19 patients in study.

	Population (N = 43)	Treated with Remdesivir (N = 9; 21%)	No Treated with Remdesivir (N = 34; 79%)
Age [mean (SD)]	69.3 (15)	61.4 (13.9)	71.4 (14.8)
Sex (M/F)	30/13	8/1	22/12
Time to onset of symptoms {days mean (SD)]	5.9 (3.86)	8.1 (3.52)	5.3 (3.78)
Disease severity 0–2 vs. disease severity 3–4 (N and %)	35 vs. 8 (81.4% vs. 18.6%)	5 vs. 4 (55.6% vs. 44.4%)	30 vs. 4 (88.2% vs. 11.8%)
Venturi mask oxygen percentage mean (SD)	0.31 (0.11)	0.35 (0.12)	0.31 (0.10)
Healed patients (N and %) vs. unhealed patients (N and %)	33 vs. 10 (76.7% vs. 23.3%)	7 vs. 2 (77.8% vs. 22.2%)	26 vs. 8 (76.4% vs. 23.6%)
High oxygen flows or NIV (N)	7	2	5
Mechanical ventilation (N)	3	3	0
Timing of sample collection relative to the onset of COVID-19 (days ± SD)	9 ± 7.2	11 ± 6.3	8 ± 7.2
N’ Asymptomatic/N’ total	6/43	0/9	6/30
N’ deaths/N’ total	5/43	1/9	4/30

**Table 4 ijms-24-14973-t004:** Comorbidities of the study population.

Comorbidities	Population (N = 43)	Treated with Remdesivir (N = 9: 21%)	Not Treated with Remdesivir (N = 34; 79%)
One or more comorbidities	28	4	24
Diabetes	3	0	3
Hypertension	20	4	16
Ischemic heart disease	3	0	3
Obesity	1	0	1
Neoplasia	8	1	7

**Table 5 ijms-24-14973-t005:** Gating strategy for the cytofluorimetric panel.

Cells Type	Gating Strategy
B lymphocytes	CD3^−^CD19^+^HLA/DR^+^
B plasmablasts	CD3^−^CD19^+^HLA/DR^+^CD38^high^
T CD8^+^ lymphocytes	CD3^+^CD8^+^
Activated effector CD8^+^ T lymphocytes	CD3^+^CD8^+^CD11b^+^
CD8^+^ memory T lymphocytes	CD3^+^CD8^+^CD11b^−^
CD4^+^ T lymphocytes	CD3^+^CD4^+^
T regulatory cells (Treg)	CD3^+^CD4^+^CD127^−^
NK T-like cells	CD3^+^CD56^+^
NK cells	CD3^−^HLA/DR^−^CD56^+^
Monocytes	CD14^+^HLA/DR^+^CD16^−^

## Data Availability

Data are available from the corresponding author upon reasonable request.

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
