# Peer review of "Changes in Lymphocyte Subpopulations after Remdesivir Therapy for COVID-19: A Brief Report"

_ijms, 2023, doi:10.3390/ijms241914973_

Round 1

Reviewer 1 Report

Comments and Suggestions for Authors

This short communication by Rosella et al emphasizes changes in the lymphocyte population in response to Remdesivir treatment. It’s a retrospective study of 43 hospitalized COVID-19 patients, including those who received Remdesivir therapy, which revealed significant alterations in lymphocyte subpopulations following treatment. B lymphocytes and plasmablasts increased significantly, cytotoxic T lymphocytes decreased, regulatory T cells reduced, and circulating monocytes increased, suggesting potential impacts on the immune response against the virus and modulation of the cytokine storm, with variations observed based on disease severity.

My comments are below:

The study included only 9 out of 43 patients who received Remdesivir therapy. It’s a small sample size, may not represent the broader population, and can introduce bias.

The study lacks strong statistical significance, while some changes in lymphocyte subpopulations were reported as significant, others had p-values close to the significance threshold (p=0.08), indicating a need for caution in interpretation. Graphs are missing for the findings.

Authors claim that significant changes were observed in the lymphocyte subpopulations before- and after- RDV therapy, however, the study lacks baseline data for lymphocyte subpopulations in COVID-19 patients before RDV therapy, making it challenging to attribute the observed changes solely to the treatment.

The authors failed to connect the observed changes in the lymphocyte population to any clinical outcome.

The study focuses on lymphocyte subpopulations and does not assess other immune system components, such as cytokines or antibody levels, hence, making it difficult to defend the study title, ‘The immunoregulatory impact of Remdesivir in COVID-19 disease.’

Comments on the Quality of English Language

English is fine but needs editing.

Author Response

Rome, 27th September 2023

Dear Editor of “International Journal of Molecular Sciences”,

first of all, my coauthors and I would like to thank You sincerely for this opportunity of cooperation, following the submission of the paper “Changes in lymphocyte subpopulation after Remdesivir in COVID-19 disease: a brief report” and its possible publication upon “International Journal of Molecular Sciences”.

We profoundly thank the reviewer for the comments and useful suggestions aimed at improving the final version of the paper.

This is a point-by-point list of changes made in the paper:

Reviewer 1

This short communication by Rosella et al emphasizes changes in the lymphocyte population in response to Remdesivir treatment. It’s a retrospective study of 43 hospitalized COVID-19 patients, including those who received Remdesivir therapy, which revealed significant alterations in lymphocyte subpopulations following treatment. B lymphocytes and plasmablasts increased significantly, cytotoxic T lymphocytes decreased, regulatory T cells reduced, and circulating monocytes increased, suggesting potential impacts on the immune response against the virus and modulation of the cytokine storm, with variations observed based on disease severity.

My comments are below:

  1. The study included only 9 out of 43 patients who received Remdesivir therapy. It’s a small sample size, may not represent the broader population, and can introduce bias.

The small sample size is due to the retrospective nature of this report. In our opinion, these data are interesting to be shown despite the small size, especially in the light of new acquisitions on other antiviral therapies (see: A molnupiravir-associated mutational signature in global SARS-CoV-2 genomes. Sanderson T, Hisner R, Donovan-Banfield I, Hartman H, Løchen A, Peacock TP, Ruis C. Nature. 2023 Sep 25. doi: 10.1038/s41586-023-06649-6).

  1. The study lacks strong statistical significance, while some changes in lymphocyte subpopulations were reported as significant, others had p-values close to the significance threshold (p=0.08), indicating a need for caution in interpretation. Graphs are missing for the findings.

We have added graphs for the findings in Figure S1,

  1. Authors claim that significant changes were observed in the lymphocyte subpopulations before- and after- RDV therapy, however, the study lacks baseline data for lymphocyte subpopulations in COVID-19 patients before RDV therapy, making it challenging to attribute the observed changes solely to the treatment.

We have rephrased the sentences. Due to the retrospective nature of this report, we have not data on lymphocyte subpopulations pre-treatment.

  1. The authors failed to connect the observed changes in the lymphocyte population to any clinical outcome.

Due to the small sample size, no evidence of lymphocyte population differences in clinical findings, such as interstitial pneumonia, has been observed. We added a sentence to state this. We have added a table with blood levels of inflammatory markers before and after RDV treatment.

  1. The study focuses on lymphocyte subpopulations and does not assess other immune system components, such as cytokines or antibody levels, hence, making it difficult to defend the study title, ‘The immunoregulatory impact of Remdesivir in COVID-19 disease.’

We have changed the title as follows: ‘Changes in lymphocyte subpopulation after Remdesivir in COVID-19 disease: a brief report’.

We thank You for your constructive critique and we hope the review process has led to an improved manuscript.

If additional changes are warranted, we will make them.

We hope that this revised version of our manuscript may now be found suitable for publication.

Sincerely,

Rossella Cianci, MD, PhD

Reviewer 2 Report

Comments and Suggestions for Authors

The paper would be benefited from addressing the following comments and questions:

1/ The title is vague and needs revision. Immunomodulatory impact is a broad term, and the scope of this study seems narrow to address the title. 

2/ Unknown abbreviations should not be used in the Abstract.

3/ Key words doesn’t reflect the content of the paper and it could be expanded.

4/ Line 62, what does it mean aim of the report? Is this a report or study? 

5/ Aim shouldn’t stand as a title. It must be in the introduction. 

6/ replace the word aim with objective

7/ why is reference needed in the aim part?

8/ after introduction, materials and method should follow, and then result

9/ Tables should be written in a chronological order. They start with Table 5. 

10/. Materials and methods are not clear. PBMC isolation, cell staining, antibodies used and concentration etc should be written carefully. Methods should be restructured carefully. Everything is mixed and hard to understand. 

11/ Conclusion doesn’t seem based on the finding and needs to be revised. 

12/ Figure 1 is not clear. Each figure should be labelled with letters and described carefully.  Whole cell population should be shown before identifying single cell and dead populations. The live/dead marker seems didn’t work. Live populations should be on the right side of the X axis.

13. The percentage of CD14.APC seems not right and the gating is not convincing. This should be redone. 

Comments on the Quality of English Language

Needs to be improved 

Author Response

Rome, 27th September 2023

Dear Editor of “International Journal of Molecular Sciences”,

first of all, my coauthors and I would like to thank You sincerely for this opportunity of cooperation, following the submission of the paper “Changes in lymphocyte subpopulation after Remdesivir in COVID-19 disease: a brief report” and its possible publication upon “International Journal of Molecular Sciences”.

We profoundly thank the reviewer for the comments and useful suggestions aimed at improving the final version of the paper.

This is a point-by-point list of changes made in the paper:

Reviewer 2

The paper would be benefited from addressing the following comments and questions:

1/ The title is vague and needs revision. Immunomodulatory impact is a broad term, and the scope of this study seems narrow to address the title.

We have changed the title as follows: ‘Changes in lymphocyte subpopulation after Remdesivir in COVID-19 disease: a brief report’.

2/ Unknown abbreviations should not be used in the Abstract.

We have modified the text, as suggested.

3/ Key words doesn’t reflect the content of the paper and it could be expanded.

We have modified the keywords, as suggested.

4/ Line 62, what does it mean aim of the report? Is this a report or study?

This is a brief report, we have modified the text, as suggested.

5/ Aim shouldn’t stand as a title. It must be in the introduction.

We have modified it accordingly

6/ replace the word aim with objective

We have modified it accordingly

7/ why is reference needed in the aim part?

We have deleted references in the aim section

8/ after introduction, materials and method should follow, and then result

The original manuscript had been structured as suggested. But in the new version, modified by the Journal staff, the two sections have been inverted. For this reason, tables do not respect chronological order. We have modified it, as suggested.

9/ Tables should be written in a chronological order. They start with Table 5.

The original manuscript had been structured as suggested. But in the new version, modified by the Journal staff, the two sections have been inverted. For this reason, tables didn’t respect chronological order. We have modified it, as suggested.

10/. Materials and methods are not clear. PBMC isolation, cell staining, antibodies used and concentration etc should be written carefully. Methods should be restructured carefully. Everything is mixed and hard to understand.

We have organized the methods in sections and reported the details of all antibodies in supplementary Table 2.

11/ Conclusion doesn’t seem based on the finding and needs to be revised.

We have revised the conclusions, as suggested

12/ Figure 1 is not clear. Each figure should be labelled with letters and described carefully.  Whole cell population should be shown before identifying single cell and dead populations. The live/dead marker seems didn’t work. Live populations should be on the right side of the X axis.

We thank the reviewer for pointing this out. We modified the Figure 1 by including letters for each panel as well as extending the figure legends. We also discriminated the whole cell population before identifying single cell and dead populations which are now plotted in a different way to better appreciate the alive from the dead cells. The live/dead dye (cat. L34975, ThermoFisher) can indeed penetrate the cell membrane in dead cells and will bind to internal proteins, resulting in very bright fluorescence.

  1. The percentage of CD14.APC seems not right and the gating is not convincing. This should be redone.

We have repeated the compensation matrix for the reported multiparameter flow cytometry analysis using FlowJO software in order to better resolve the HLA-DR high cells within the CD14/APC+ cells. We have indeed generated new plots and given an example of gating in Figure 1 (see “N” panel). Nonetheless, the new percentage values for HLA-DR high classical monocytes are similar to our previous analysis and these data are in line with our previous results.

We thank You for your constructive critique and we hope the review process has led to an improved manuscript.

If additional changes are warranted, we will make them.

We hope that this revised version of our manuscript may now be found suitable for publication.

Sincerely,

Rossella Cianci, MD, PhD

Reviewer 3 Report

Comments and Suggestions for Authors

Dear Author,

This is an interesting piece of work in an extremely important area, which raises questions that are useful for prospective studies. The authors present significant changes in the lymphocyte subpopulations analyzed before- and after Remdesivir (RDV) therapy.  However, participants in this study represent an extremely heterogeneous group relative to disease progression and comorbidities.  The mechanism of alterations in lymphocytes subpopulations, especially in severe stages of COVID-19 is very complex and there are several possible factors that may contribute to this phenomenon. The majority of individuals with weakened immune systems and patients with pre-existing comorbidities are more vulnerable to severe SARS-CoV-2 infection. There are alterations in lymphocytes subpopulations in patients with neoplasma, hypertension (HT), obesity, specifically in type 1 diabetes (T1DM) and type 2 diabetes (T2DM). The activity of these lymphocyte subpopulations can vary depending on disease stage and overall immune response of the individual. In individuals with HT, obesity, T1DM, especially in T2DM, there is often a low-grade inflammation, which can lead to changes in level and activity of various pro-inflammatory cytokines, which can influence lymphocyte function and differentiation. Further research is needed with more homogenous groups (inclusive SARS-CoV-2 positive/negative groups in different stages of COVID-19) and a larger sample size to understand the mechanisms underlying the specific changes in lymphocyte subpopulations and the factors that may influence results of clinical research.

 Issues that the author should consider in order to strengthen the manuscript are as follows:

(i)              Line 149: The study had previously (please explain) received appropriate institutional ethical approvals by Ethical Committees. Ethic committee data and protocols must be added;

(ii)             Please note that the structure of MS should also include Discussion/Conclusions and Future directions according to IJMS recommendations;

(iii)           The main problem in this study is the limited sample size in combination with the high number of markers and comorbidities. Moreover, the analysis has technical deficits: statistical analysis or cross-variable comparisons, and controls without comorbidities are lacking. As stated in the recommendations, clinical characteristics of the subjects, comorbidities, time from symptom onset, and disease severity are important in order to provide an adequate and well-controlled study in a related disease area. I would recommend adding the timing of serum collection relative to the onset of COVID-19 disease and clinical characterization of both control group and COVID-19 diagnosed patients or an explanation. 

(iv)            In Discussion/Conclusions the author overinterpreted the results and I would suggest that the author add more information and compare the results they obtained with other articles. Please highlight the limitations and implications for researchers carrying out future studies in the same field, and for public health practice.

Comments on the Quality of English Language

The Quality of English is acceptable

Author Response

Rome, 27th September 2023

Dear Editor of “International Journal of Molecular Sciences”,

first of all, my coauthors and I would like to thank You sincerely for this opportunity of cooperation, following the submission of the paper “Changes in lymphocyte subpopulation after Remdesivir in COVID-19 disease: a brief report” and its possible publication upon “International Journal of Molecular Sciences”.

We profoundly thank the reviewer for the comments and useful suggestions aimed at improving the final version of the paper.

This is a point-by-point list of changes made in the paper:

Reviewer 3

Dear Author,

This is an interesting piece of work in an extremely important area, which raises questions that are useful for prospective studies. The authors present significant changes in the lymphocyte subpopulations analyzed before- and after Remdesivir (RDV) therapy.  However, participants in this study represent an extremely heterogeneous group relative to disease progression and comorbidities.  The mechanism of alterations in lymphocytes subpopulations, especially in severe stages of COVID-19 is very complex and there are several possible factors that may contribute to this phenomenon. The majority of individuals with weakened immune systems and patients with pre-existing comorbidities are more vulnerable to severe SARS-CoV-2 infection. There are alterations in lymphocytes subpopulations in patients with neoplasma, hypertension (HT), obesity, specifically in type 1 diabetes (T1DM) and type 2 diabetes (T2DM). The activity of these lymphocyte subpopulations can vary depending on disease stage and overall immune response of the individual. In individuals with HT, obesity, T1DM, especially in T2DM, there is often a low-grade inflammation, which can lead to changes in level and activity of various pro-inflammatory cytokines, which can influence lymphocyte function and differentiation. Further research is needed with more homogenous groups (inclusive SARS-CoV-2 positive/negative groups in different stages of COVID-19) and a larger sample size to understand the mechanisms underlying the specific changes in lymphocyte subpopulations and the factors that may influence results of clinical research.

 Issues that the author should consider in order to strengthen the manuscript are as follows:

  • Line 149: The study had previously (please explain) received appropriate institutional ethical approvals by Ethical Committees. Ethic committee data and protocols must be added;

We have added Ethic committee data and protocols

(ii)             Please note that the structure of MS should also include Discussion/Conclusions and Future directions according to IJMS recommendations;

We have added a section on future directions

(iii)           The main problem in this study is the limited sample size in combination with the high number of markers and comorbidities. Moreover, the analysis has technical deficits: statistical analysis or cross-variable comparisons, and controls without comorbidities are lacking. As stated in the recommendations, clinical characteristics of the subjects, comorbidities, time from symptom onset, and disease severity are important in order to provide an adequate and well-controlled study in a related disease area. I would recommend adding the timing of serum collection relative to the onset of COVID-19 disease and clinical characterization of both control group and COVID-19 diagnosed patients or an explanation.

We added the timing of serum collection relative to the onset of COVID-19 disease in Table 1.

(iv)            In Discussion/Conclusions the author overinterpreted the results and I would suggest that the author add more information and compare the results they obtained with other articles. Please highlight the limitations and implications for researchers carrying out future studies in the same field, and for public health practice.

We thank the reviewer for pointing this out. We added a section on future directions and highlighting the limitations of this study.  A comparison with the results of previous studies is also included.

We thank You for your constructive critique and we hope the review process has led to an improved manuscript.

If additional changes are warranted, we will make them.

We hope that this revised version of our manuscript may now be found suitable for publication.

Sincerely,

Rossella Cianci, MD, PhD

Round 2

Reviewer 1 Report

Comments and Suggestions for Authors

Although the authors have made an effort to address all the comments provided, I find that the overall quality of the study does not meet the standards expected by the journal. The authors claimed data to be interesting despite the small size, especially in the light of new acquisitions on other antiviral therapies (see: A molnupiravir-associated mutational signature in global SARS-CoV-2 genomes. Sanderson T, Hisner R, Donovan-Banfield I, Hartman H, Løchen A, Peacock TP, Ruis C. Nature. 2023 Sep 25. doi: 10.1038/s41586-023-06649-6). Can authors explain the correlation between both studies? 

It is imperative to have access to baseline data regarding lymphocyte subpopulations in COVID-19 patients prior to receiving Remdesivir therapy. Without this crucial baseline information, it becomes exceedingly challenging to draw meaningful interpretations or conclusions from the study's results.

Lastly, in light of the limitations inherent in the study's findings, it is indeed challenging to arrive at any definitive conclusions. Furthermore, the findings are not adequately discussed, leaving a significant gap in the understanding of the research's implications and potential impact.

Comments on the Quality of English Language

English quality is ok.

Author Response

Rome, 02nd October 2023

Dear Editor of “International Journal of Molecular Sciences”,

First of all, my coauthors and I would like to thank You sincerely for this opportunity of cooperation, following the submission of the paper “Changes in lymphocyte subpopulation after Remdesivir in COVID-19 disease: a brief report” and its possible publication upon “International Journal of Molecular Sciences”.

We profoundly thank the reviewers for the comment of the final version of the paper. Thanks to the reviewers’ constructive and detailed comments, we feel that our manuscript is now clearer and easier to understand, constituting a better context for the theme to be discussed. 

This is a list of changes requested:

Reviewer 1

Although the authors have made an effort to address all the comments provided, I find that the overall quality of the study does not meet the standards expected by the journal. The authors claimed data to be interesting despite the small size, especially in the light of new acquisitions on other antiviral therapies (see: A molnupiravir-associated mutational signature in global SARS-CoV-2 genomes. Sanderson T, Hisner R, Donovan-Banfield I, Hartman H, Løchen A, Peacock TP, Ruis C. Nature. 2023 Sep 25. doi: 10.1038/s41586-023-06649-6). Can authors explain the correlation between both studies?

Sanderson et al. have shown that an antiviral drug, such as Molnupiravir, widely used against SARS-CoV-2, induces a mutational spectrum, with preferred nucleotide contexts, in this virus. Our sentence aimed to highlight that the selective pressure of the antiviral therapies combined with the natural rapid onset of new SARS-CoV-2 variant strains need to search for effective antiviral therapies. Moreover, we conclude that ’Further larger prospective researches are needed to identify the specific contribution to immune modulation by this antiviral treatment, its link with clinical outcome and cytokine production in more homogeneous subject subgroups’. We did not correlate the two studies.

It is imperative to have access to baseline data regarding lymphocyte subpopulations in COVID-19 patients prior to receiving Remdesivir therapy. Without this crucial baseline information, it becomes exceedingly challenging to draw meaningful interpretations or conclusions from the study's results.

Lastly, in light of the limitations inherent in the study's findings, it is indeed challenging to arrive at any definitive conclusions. Furthermore, the findings are not adequately discussed, leaving a significant gap in the understanding of the research's implications and potential impact.

Unfortunately, as specified in response to the first report (point 3 of author’s response), due to the retrospective nature of this report, we do not have data on lymphocyte subpopulations pre-treatment. This puts us in the position of being unable of addressing the reviewer’s point.

We have addressed the limitations of our study in the discussion by making explicit that these are due to the retrospective nature of this analysis.

Kind Regards,

Rossella Cianci, MD, PhD

Reviewer 2 Report

Comments and Suggestions for Authors

The second revision is significantly improved and comments are addressed promptly. One minor comment to be considered is: The ' future direction' section can be included in the 'conclusion' section but in a much brief way. 

Author Response

Rome, 2nd October 2023

Dear Editor of “International Journal of Molecular Sciences”,

first of all, my coauthors and I would like to thank You sincerely for this opportunity of cooperation, following the submission of the paper “Changes in lymphocyte subpopulation after Remdesivir in COVID-19 disease: a brief report” and its possible publication upon “International Journal of Molecular Sciences”.

We profoundly thank the reviewer 2 for the comment of the final version of the paper.

Reviewer 2

The second revision is significantly improved and comments are addressed promptly. One minor comment to be considered is: The ' future direction' section can be included in the 'conclusion' section but in a much brief way.

Thank you for your suggestion.

But, we have previously added this separate section on future directions, following a specific request of Reviewer 3.

We thank the reviewer for the constructive critique and we hope the review process has led to an improved paper.

Kind regards,

Rossella Cianci, MD, PhD

Reviewer 3 Report

Comments and Suggestions for Authors

I am satisfied with the author´s responses to my issues raised in my initial review.

Author Response

Rome, 2nd October 2023

Dear Editor of “International Journal of Molecular Sciences”,

First of all, my coauthors and I would like to thank You sincerely for this opportunity of cooperation, following the submission of the paper “Changes in lymphocyte subpopulation after Remdesivir in COVID-19 disease: a brief report” and its possible publication upon “International Journal of Molecular Sciences”.

We profoundly thank the reviewer 3 for the comment of the final version of the paper.

Reviewer 3

I am satisfied with the author´s responses to my issues raised in my initial review.

Thank you for your comment.

We thank the reviewer for the constructive critique and we hope the review process has led to an improved paper.

Kind regards,

Rossella Cianci, MD, PhD

Round 3

Reviewer 1 Report

Comments and Suggestions for Authors

Can be accepted.